# *"Tell them you smoke, you'll get more breaks"*: a qualitative study of occupational and social contexts of young adult smoking in Scotland

Hannah Delaney,[1] Andrew MacGregor,[1] Amanda Amos[2]

[1]ScotCen Social Research, Edinburgh, UK
[2]Usher Institute of Population Health Sciences and Informatics, The University of Edinburgh, Edinburgh, UK

**Correspondence to**
Professor Amanda Amos;
Amanda.Amos@ed.ac.uk

## ABSTRACT

**Objective** To explore young adults' perceptions and experiences of smoking and their smoking trajectories in the context of their social and occupational histories and transitions, in a country with advanced tobacco control.
**Design** Indepth qualitative interviews using day and life grids to explore participants' smoking behaviour and trajectories in relation to their educational, occupational and social histories and transitions.
**Setting** Scotland.
**Participants** Fifteen ever-smokers aged 20–24 years old in 2016–2017.
**Results** Participants had varied and complex educational/employment histories. Becoming and/or remaining a smoker was often related to social context and educational/occupational transitions. In several contexts smoking and becoming a smoker had perceived benefits. These included getting work breaks and dealing with stress and boredom, which were common in the low-paid, unskilled jobs undertaken by participants. In some social contexts smoking was used as a marker of time out and sociability.
**Conclusions** The findings indicate that while increased tobacco control, including smokefree policies, and social disapproval of smoking discourage smoking uptake and increase motivations to quit among young adults, in some social and occupational contexts smoking still has perceived benefits. This finding helps explain why smoking uptake continues into the mid-20s. It also highlights the importance of policies that reduce the perceived desirability of smoking and that create more positive working environments for young adults which address the types of working hours and conditions that may encourage smoking.

## INTRODUCTION

Reducing smoking uptake is a key goal of tobacco control strategies. In the UK recent governmental action has included reducing tobacco promotion (banning point-of-sale displays; standardised packaging), reducing cigarettes' affordability (taxation, banning small packs) and availability (increasing age of sale), increasing awareness of health risks (media campaigns, health warnings), and

## Strengths and limitations of this study

► This is one of the few qualitative studies to explore smoking uptake and trajectories in British young adults.
► We recruited young adults aged 20–24 years old with diverse smoking histories, and educational and occupational trajectories.
► The indepth interviews used both day and life grids to explore current and previous smoking patterns in relation to social, occupational and educational contexts.
► As has been found in previous studies, recruiting a purposive sample in this age group was challenging. The time between taking part in the Scottish Health Surveys and the qualitative interviews was 1–3 years, increasing the likelihood of changed addresses and phone numbers.

reducing the social acceptability of smoking (smokefree public places and cars).[1] These measures have significantly impacted on youth smoking. Smoking prevalence in Scottish 15-year-olds halved between 2008 and 2015, from 15% to 7%.[2]

However, as in many countries,[3] smoking uptake in the UK continues into the mid-20s. Also the decline in smoking among those aged 16–24 years old has been less than in younger age groups, from 28% in 2008 to 21% in 2016 in Scotland.[4] This age group is highlighted in the Scottish Tobacco Control Strategy[5] and in the English Tobacco Plan[6] as being of concern, as over a third of smokers aged 16–24 years old started smoking at age 16 or over.[5] Similarly, a US longitudinal study found that 18% of ever-smokers under 30 years started smoking between 18 and 21,[7] and in the European Union 41% of ever-smokers started regular (weekly) smoking between 18 and 25.[3] Understanding the smoking beliefs, behaviour and social contexts of young adults is vital for developing effective strategies to reduce smoking in this key age group.

Young adulthood can be a time of increased autonomy and freedom to explore different identities and behaviours before more stable roles and responsibilities in later adult life.[8] Young adults often move in and out of smoking,[9] and health behaviours can be taken up, consolidated or abandoned.[10 11] This period of fluidity presents an opportunity to prevent never smokers or those who regard themselves as 'social' smokers from becoming regular smokers, and to encourage smokers to quit.[12–14] Qualitative studies have found that identity construction and presentation of self, which Goffman[15] conceptualised as staging a performance that is expected in certain situations and that is credited with desired attributes by other actors in that context, are important in understanding young adults' smoking.[16–21] The transitions of young adulthood, which traditionally involve leaving school, leaving the parental home, taking up full-time employment, starting cohabitation and having children,[22] also shape smoking behaviour. Wiltshire et al's[23] study of Scottish aged 16–19 years old highlighted the impact of transitions from school to work, further education or unemployment on becoming and staying a smoker. Smoking was perceived by smokers as a lubricant for social relations and a marker of an acceptable identity in new occupational or social contexts, which reinforced and increased smoking. Studies in the USA, New Zealand and UK have found that smoking and drinking are highly associated, particularly among young adults who typically spend more time than older adults socialising with friends in bars, clubs and at parties.[14 20 21 23 24]

It was expected that the UK's smokefree legislation might particularly impact on this age group through reducing opportunities to smoke in educational, occupational and leisure settings, requiring smokers to go outside to smoke, thereby disrupting the perceived social role and value of smoking in these contexts.[23] This change has added importance for smokers who despite often high consumption levels, usually when drinking alcohol, do not regard themselves as 'proper' smokers and risk becoming regular smokers.[12 13 21 25 26] Rooke et al's[27] qualitative study of the English smokefree legislation found that it was accepted by young adult smokers as it represented a continuation of smoking denormalisation processes that had characterised their lives. However, smoking remained for them an integral activity of the night-time economy, and a marker of pleasure and sociability. This echoes findings from New Zealand, where smoking among young adults has moved to the 'liminal' areas outside bars where smoking is permitted.[21 28] Rooke et al's[27] study was undertaken 10 years ago following the legislation. It is important to gain a contemporary understanding of how smoking is perceived by young adult smokers, in particular how the now embedded smoking restrictions and continued denormalisation of smoking are shaping smoking behaviour and trajectories, not least at times of significant life transitions.

This qualitative study is part of a larger study investigating smoking patterns and trends in young adults aged 16–24 years old in Scotland. It aimed to explore, through indepth interviews, young adults' perceptions and experiences of smoking, and smoking trajectories, in the context of their social and occupational histories/transitions. It also aimed to increase our understanding of why, in a country with the most advanced tobacco control in Europe,[29] smoking uptake continues into the mid-20s.

## METHODS

This paper presents the qualitative findings from a mixed-methods study involving (1) secondary analysis of the 2012–2015 annual Scottish Health Surveys (SHeS)[4] examining smoking young adults aged 16–24 years old by sociodemographic factors and (2) qualitative interviews with purposively selected 2014–2015 survey respondents in 2016–2017. SHeS is Scotland's most robust national data set on smoking.[4]

Purposive sampling of SHeS respondents aged 16–24 years old was undertaken to recruit participants who had consented to be recontacted for follow-up research. Preliminary analysis (SHeS data set, 2012–2015) of the 2012–2015 survey data showed that smoking prevalence varied by age (16% in those aged 16–19 years old vs 26% in those aged 20–24 years old) and economic activity, with those aged 20–24 years old in full-time education reporting the lowest smoking prevalence (20%), compared with 25% in the employed and 42% in the NEET (Not in Education, Employment, Training) categories. The sampling aimed to recruit a diverse sample with differing smoking and occupational histories using the following criteria: smoking status (ever-smoked), age at qualitative interview (20–24), gender and economic activity. Invitation letters, information sheets and consent forms were sent to 85 eligible SHeS respondents. Fifteen returned consent forms and were interviewed in December 2016–April 2017. Only 13 individuals refused to be interviewed. Nine invitations were returned as 'no longer at this address'. In 48 cases there was no response to invitation letters and up to five calls and texts. Participants received a £20 high street voucher.

The interviews were conducted by two female experienced qualitative researchers in participants' homes (interview guide in online supplementary file 1). They were digitally recorded and lasted 1–2 hours. Interviews explored participants' current smoking behaviour and history, occupational and social transitions (eg, leaving school, unemployment/employment, new social contexts/networks), and the perceived influence of these transitions and contexts on their smoking. The interview used two adapted versions of the 'life grid' (online supplementary file 2). Versions of the life grid have been used in qualitative studies of smoking to collect retrospective smoking histories[30] and current smoking behaviour across the course of a day.[31 32] The life grid recorded structured data on a timeline (before 16 years and each subsequent year) of the participant's work/education, social and smoking history. Key changes and transitions

were recorded for each year. The day grid recorded participants' daily routines and smoking behaviour for each hour of a typical and atypical (eg, weekend) day. The grids were completed as a joint endeavour between the interviewer and the participant. They helped build rapport and were often returned to during the interview, allowing the interviewer and the participant to reflexively refocus and/or elaborate on themes.

The interviews were transcribed and entered into NVivo V.10 to facilitate data management. The transcripts were read in conjunction with the grids to contextualise smoking behaviour and trajectories. Data were analysed thematically, informed by Braun and Clarke's[33] phases of thematic analysis. The initial analysis involved familiarisation; transcripts were read and reread by the coauthors and emergent themes discussed. Codes were systematically compared to identify cross-cutting themes and highlight common experiences, as well as differing views. The coding framework was further refined into key themes. While we cannot say that data saturation was reached, given the diverse sample, the participants did discuss similar experiences and their accounts included several overlapping themes. Where quotations are used in the Results section, participants are identified by a pseudonym, age in years and smoking status.

### Patient and public involvement
No patient or public were involved.

### RESULTS
#### Participants' employment and smoking status
Participants tended to move from education or unemployment (NEET) into employment between the SHeS and the qualitative interview (table 1). Two participants were now in full-time education, 12 were employed (10 full-time, 2 part-time) and 1 was still unemployed.

The interviews revealed varied and complex employment histories, with most of those in employment having

**Table 1** Sample characteristics at the time of the SHeS and the qualitative interview

| | SHeS (2014–2015) | | | Qualitative interview (2016–2017) | | |
|---|---|---|---|---|---|---|
| | Male | Female | Total | Male | Female | Total |
| Current smoker | 7 | 6 | 13 | 6 | 3 | 9 |
| Ex-regular | 1 | – | 1 | 2 | 2 | 4 |
| Ex-occasional | 1 | – | 1 | 1 | 1 | 2 |
| Education | 2 | 2 | 4 | – | 2 | 2 |
| Employment | 6 | 2 | 8 | 9 | 3 | 12 |
| NEET | 1 | 2 | 3 | – | 1 | 1 |
| Total | 9 | 6 | 15 | 9 | 6 | 15 |

NEET, Not in Education, Employment, Training; SHeS, Scottish Health Surveys.

had several jobs since leaving school. For instance, since leaving school at 17, Rachel (22, ex-smoker) had had five jobs in three different sectors (hospitality, retail, hairdressing). Participants' current employment sectors included retail, hospitality (eg, bars, restaurants), skilled manual labour (eg, painting, roofing) and administration/services (eg, call centres), with shift work, part-time and zero-hour contracts common. Nine participants had attended college or university and two had been apprentices. Only three participants had followed the 'traditional' route of university/college or apprenticeship followed by full-time employment. Six participants had at some point been unemployed for at least several weeks.

Nine participants were current smokers. Four of the six ex-smokers had quit since the survey (table 1). Eleven had started smoking before 16, mostly trying their first cigarette between 13 and 15 in social situations (table 2). Four participants started smoking later, one at 17 and three at 18. This was typically associated with socialising at parties or clubs when drinking, and starting university/college or a new job. Smoking consumption varied considerably from 3 to 30 cigarettes a day.

While most had smoked for several years, only two described trajectories of increasing consumption since starting smoking. For other participants, irrespective of age of first trying a cigarette, accounts included at least one attempt to quit (often short-lived), and significant increases and decreases in consumption. Of the six ex-smokers, two were occasional and four were regular smokers before quitting. Some articulated clear reasons for deciding to quit, such as significant events (eg, relative's death from a smoking-related illness), while others had more general reasons including no longer finding smoking appealing, wanting to improve their health and finances, and their peer group no longer comprising smokers.

Three themes emerged as important in influencing smoking behaviours and trajectories: occupational role and context, social context, and domestic context.

#### Occupational smoking
Participants' accounts highlighted that there still appear to be opportunities to smoke, and even to start smoking, at work. As workplaces adhere to smokefree legislation, smokers take breaks outside the office or other enclosed environments. Due to the relatively transient nature of occupations in this age group, temporary jobs such as hospitality, retail and shift work were common. Many participants described these work contexts as encouraging them to move from being non-smokers or social smokers (only borrowing cigarettes from friends) to regular smokers (buying their own cigarettes).

Participants described the functional and social benefits of taking up smoking at work, on the one hand to get more breaks and relieve boredom and stress, and on the other as a means of socialising with colleagues. For instance, Sarah (22, smoker) started waitressing at 18 to save money before going travelling and was advised: *"Tell*

**Table 2** Smoking history of interview respondents

| Name* | Age | Age first smoked | Smoking history | Current smoking status |
|---|---|---|---|---|
| Adam | 23 | 14 | Quit for 3 months at 19 | Current smoker (30/day) |
| Rachel | 22 | 13 | Quit at 16, relapsed at 19, quit at 20 | Ex-regular smoker |
| Tom | 20 | 10 | Quit at 14, relapsed at 16, cut down recently | Current smoker (3–5/day) |
| Sarah | 22 | 18 | Quit for 3 months at 19 | Current smoker (5–7/day) |
| Louise | 22 | 18 | Trying to cut down and quit | Current smoker (10/day) |
| Daniel | 23 | 17 | Quit attempt at 20 | Current smoker (15–20/day) |
| Jamie | 24 | 12 | Quit for 6 months at 19, cut down with e-cigarette at 22 | Ex-regular smoker |
| Duncan | 23 | 12 | Quit for 1 week at 18 | Current smoker (12/day) |
| Peter | 24 | <10 | Quit at 17 | Ex-occasional smoker |
| Kate | 23 | 13 | Quit attempt at 20 postpregnancy | Current smoker (+6/day) and uses e-cigarette |
| Rob | 24 | 17 | Quit for 2 days at 21, cut down/quit at 24 | Ex-regular smoker and uses e-cigarette |
| Chris | 23 | 14 | Quit for 6 months at 19 | Current smoker (4–5/day) |
| Stephen | 23 | 15 | Cut down recently | Current smoker (10/day) |
| Alison | 21 | 18 | Very low occasional smoker | Ex-occasional smoker |
| Helen | 23 | 12/13 | Quit for 3 months at 21, cut down recently | Current smoker (8–10/day) |

*All names are pseudonyms.

them you smoke. You'll get more breaks." She explained how this led to smoking initiation: *"you don't even want a cigarette, but you would just go out and maybe walk around for a bit, and then it got to the point where we would just start having cigarettes."* Similarly, Jamie (24, ex-smoker) had first tried a cigarette at age 12 but had no interest after that, until he started working at a hotel when 16 and became a regular smoker:

> At the hotel you weren't allowed a break unless you were a smoker, so I used to go out with the smokers for like a 5 min break, multiple times a day, and that's when I'd have one, so that they didn't know that I wasn't a smoker…you can't be kind of on a break if you're not having a fag, so, if the manager comes out, you need to be holding one!

Others described their daily smoking pattern at work as being deeply ingrained, often as a stress relief: *"a quick moment of solace to go and bolster yourself"* (Chris, 23, smoker), or as a habit that coincided with breaks. Tom's (20, smoker) smoking consumption was determined by the length of breaks during his call centre shift: *"in the half hour break, you used to have two cigarettes…and then my 15 min I used to have one."* Relief from boredom also encouraged smoking: *"It turned into more of a dependency when I was at work and I found myself sort of clock watching, waiting to get to my break"* (Rob, 24, ex-smoker). Similarly, Sarah (22, smoker) reported *"smoking more than ever"* due to boredom in her call centre job.

For those in more permanent employment, the perceived benefit of smoking seemed to become rooted in the social aspect of taking work breaks with colleagues.

Daniel (23, smoker) resisted smoking throughout school; however, on starting his first office job at 18, he bought a packet of cigarettes so that he could join his team for a cigarette break. He now smoked 15–20 a day:

> I only had like a couple for the first few weeks…I didn't actually enjoy it when I first started it at all. It was just a case of now and again just going out for the sake of it, or if I'm getting really ticked off at work. But then, after about a few weeks, it was a case of smoking the 10, and then all of a sudden it's the 20.

In contrast to the perceived benefits of smoking at work, others described how they moderated their smoking due to concerns around professional competency, acceptability and demeanour. Stephen (23, smoker) reduced his smoking from over 30 to 10–15 a day while training for a fitness test as part of his Navy application. Sarah (22, smoker) reported never smoking during her student nursing placements, as she did not want to present herself to patients as a smoker:

> I really didn't like smoking on placements: coming back and dealing with patients, just stinking of fags. I just don't think it's that professional.

Tom (20, smoker) explained that as his colleagues did not smoke, he didn't want to *"come across like smelly,"* and so masked the smell to present himself in an acceptable manner to them and customers at the cinema where he worked:

> Like I can smell it off me sometimes, so I have like a can o' deodorant in the locker room. Chewing gum,

mints. Something like that basically…coz I'm dealing wi' customers an' that…so it's nice to be…like have fresh breath, when I'm talking to people, come across a certain way…presentable to the public basically.

Similarly, several participants described how the perceived social benefits of smoking at work could diminish following changes in social groups due to occupational or educational transitions, resulting in reduced smoking or quitting. Jamie's (24, ex-smoker) first quit attempt was during his first 6 months of university due to his new peers being non-smokers. Similarly, Kate (23, ex-smoker) smoked 20 cigarettes a day, but this reduced to 6 when she started a college course with a new group of non-smoking peers:

> Different course, different people, most of the people on this course don't actually smoke. So it's not very nice just having to go out by myself for a fag. You want to quickly smoke your fag rather than take your time and talk to people so it became about 6 a day on this course.

### Smoking and drinking

All participants reported being exposed to smoking when out socially, particularly outside bars and clubs. Smoking increased when drinking alcohol, with several reporting that currently or previously they only smoked when drinking at parties, bars or clubs.

> I smoke quite a lot if I'm out drinking or whatever, but, apart from that, not really. (Duncan, 24, smoker)

Several participants, including three of the four who started after the age of 16, traced starting smoking to 'social' smoking while drinking. Sarah (22, smoker) recalled how her regular smoking began at 18 during the summer when she was drinking and socialising a lot. Similarly, Louise (22, smoker) explained how social smoking while drinking developed into regular smoking:

> It was really just a gradual kind of increase, it wasn't any kind of event that said I'm going to smoke more today. I think going out and going to clubs and having a drink and everything does kind of make you smoke more.

Other participants discussed their quit attempts and highlighted drinking and socialising as triggering relapse. Tom (20, smoker) talked about struggling to avoid the *"fall back"* into smoking when drinking. Others discussed similar situations, with some cutting down or stopping smoking during the week but *"weekend wise probably still sticking at the same [smoking levels]"* (Rob, 24, ex-smoker).

### Smoking and living circumstances

Participants talked about phases when their smoking increased or decreased due to changed social and living circumstances, reflecting the often-transient nature of their lives. Seven currently lived independently in rented accommodation and others had had periods away from home, for example, at university. They discussed changes in their smoking levels due to moving away from home for the first time and the freedom of having their own space and money, living in other countries where tobacco was more or less accessible and affordable, and the smoking status of who they lived with.

For most, the transition to independent living occurred during their late teens. This coincided with increased socialising and drinking, changes in social circles and more control over their finances, all of which were associated with increased smoking. Rachel (22, ex-smoker) explained that she *"went a bit wild"* when she first left home and her smoking increased:

> I'd just split up with my boyfriend of 3 years…so and because I was out of the house [parent's home]… there was nobody telling me that I couldn't so I just… think I done it [smoked] because I could.

Stephen (23, smoker) also explained that when he moved to university:

> My parents gave me a budget of £200 a month and I was meant to get a job but, I don't know, I just didn't prioritise things like that. So my £200 budget, I used that to buy cigarettes quite a lot.

Several participants had a history of transient living circumstances and perceived that changes in their location had affected their smoking. For instance, Chris (23, smoker) reported that his smoking increased when he temporarily relocated to Germany for work, as tobacco was cheaper there. In contrast, Sarah (22, smoker) reported that her smoking decreased when travelling abroad on her 'gap' year due to a changed routine and tobacco being harder to access.

For others, changes in who they lived with affected their smoking. For instance, Jamie (24, ex-smoker) had quit smoking for 6 months but *"things took a turn"* when a flatmate who smoked moved in. He started to go for a cigarette with this flatmate and then *"kinda got the taste back again…and that's when I kinda started buying them again."* Participants who had or were living with partners also reported changes in smoking consumption due to their relationships. Kate (23, ex-smoker) explained that smoking had played a big part in her relationship with her (now) ex-partner, as it was something that they would do together. Her current partner did not smoke, and consequently she had stopped smoking:

> I think it's because like I'm a smoker and my partner is a non-smoker and I was like, you know, it must be so gross…So I was just kind of like…it doesn't seem fair.

Similarly, Chris (23, smoker) smoked less around his partner due to health concerns:

> He's actually got like not very strong lungs, so I don't tend to smoke around him, and actually…probably having been with him for the last few years maybe my smoking has gone down just because I know that it

can affect him more…He'll cough, and it'll make me feel bad, so I won't do it!

Participants who had only lived with their parent(s) attributed changes in their smoking status or consumption to factors other than their living circumstances, such as changes in social and employment contexts, as previously described.

## DISCUSSION

This paper reports the findings of indepth interviews with young adults which explored their smoking behaviour and trajectories, in the context of their social and occupational histories since leaving school. It aimed to increase our understanding of why, in a country with strong tobacco control where smoking is increasingly denormalised, smoking uptake continues into the mid-20s.

The study shows how young adulthood is a period of considerable flux and transition, where becoming and/or remaining a smoker is often related to social context and the nature of educational and occupational transitions. As was found in research conducted with young adults 10 years ago shortly after the smokefree legislation,[27] participants accepted that their smoking behaviour was subject to both formal and informal social controls. These included legal restrictions on smoking in public places and presenting a socially acceptable image in certain professional and social contexts. Participants talked about, without question, going outside to smoke when at work or pubs and clubs, and not exposing non-smoking partners and friends to their smoking in certain social contexts. The need to manage their smoker identity in different contexts was most marked in accounts of smoking not being an acceptable part of the performance of a professional self in jobs involving contact with the public, such as nursing and the hospitality business. This reaction reflects the increasingly negative social climate around smoking in the UK and countries such as New Zealand,[22 34] with smoking being stigmatised as a marker of low social status,[31 35 36] what Goffman[15] has described as a spoilt, polluted identity. In these contexts, young adult smokers' appropriate presentation of self appeared to be paramount. While such social controls and meanings could discourage smoking uptake, reduce consumption and increase motivations to quit, there were other contexts where smoking was perceived more positively.

Despite smoking restrictions in pubs and clubs, smoking in these social contexts was construed by many as being sociable and inherent to relaxing with friends and drinking alcohol. This could create conflicting self-identities for participants: between the weekday 'structured/controlled' smoker and the weekend 'uncontrolled' social smoker. As has been found previously, in these contexts, smoking not only increased, which could lead to sustained increased consumption, but could encourage young adults who had not smoked when at school to smoke their first cigarettes,[21 24] perhaps, as has been found in other studies, in the belief that 'social smoking' would not lead to regular smoking or that social smokers could deny their smoker identity.[13 37–40] These were also common in contexts where quit attempts failed.

Perhaps the most unexpected finding was that young adults taking up jobs with particular employment conditions could lead to increased smoking, to deal with stress and boredom, and to smoking initiation. Several participants described how in certain occupational contexts, notably the hospitality industry and continuous demand jobs in call centres, being a smoker carried the significant benefit of short breaks, and in some cases was the only way of getting breaks. Such accounts were rare in presmokefree studies where taking up smoking on transitioning from school to work was attributed more to 'fitting in' with new colleagues.[23] This finding may partly reflect the more transitional and precarious nature of contemporary young adults' lives, where changes in the employment market during the recession have led to more part-time working and low-paid jobs in this age group.[41 42] Also more full-time students in the UK are undertaking part-time or temporary jobs to fund their studies.[43] These tend to be in low-paid, low-skilled occupations which generally have poorer rights, which can lead to 'role overload' where transitions into adult roles can create stressful demands.[11] Thus while the smokefree legislation and declining smoking prevalence mean that young adults are less exposed to pro-smoking influences at work, in some contexts pressures to smoke or start smoking remain.

Finally, the study reconfirmed the importance of peer influence on smoking trajectories in young adulthood across all social, educational and occupational contexts and transitions.[20 23 27 38] Who participants socialised, worked and/or lived with was perceived as impacting on their smoking status and behaviour. Leaving school for college or employment and leaving home and gaining more independence go hand in hand with forging new social networks,[23 44] which may have different smoking norms which promote or inhibit smoking.

The main limitation of this study was recruitment. As has been found in previous studies, recruiting young adults into research is difficult as they frequently change addresses given their often transient lives and living circumstances.[14] Additionally, although most SHeS households provided a phone number, this was often not the young person's phone number; the time between taking part in SHeS and the qualitative interviews was 1–3 years, increasing the likelihood of changed addresses and phone numbers. These qualitative data from a small purposive sample cannot be generalised to the Scottish population.

While it was not possible to include all patterns of smoking and occupational trajectories, the participants had diverse smoking histories (eg, started smoking before/after leaving school, increased and/or decreased their smoking, not/tried to quit) and educational/occupational histories. This diversity was a strength of the

study particularly given the above acknowledged challenges of recruiting young adults to research studies. This diversity generated potentially important contemporary insights about how such transitions and contexts can affect smoking. It is possible that other young adults may have different experiences, particularly those from more disadvantaged backgrounds where they may experience fewer transitions and opportunities. There is also a need to explore how young adults experiencing similar transitions and influences resist smoking. Further studies should explore these issues with young adults in differing social and economic circumstances.

In conclusion, this study indicates that despite increased tobacco control policies and social disapproval of smoking, in some social and occupational contexts smoking and having a smoking identity still have perceived positive benefits for young adults. Smoking can be used to deal with occupational pressures through getting 'smoking breaks' and/or perceived relief from stress and boredom which are inherent in many low-paid, unskilled jobs undertaken by young adults. In addition, in some social contexts, smoking is used as a marker of time out and sociability. This perhaps helps explain why the decline in young adults' smoking is slower than that in younger adolescents where the more restrictive contexts of school and home environments are barriers to smoking uptake.[23] Our findings show how leaving these environments for more transient, independent lives can facilitate or discourage smoking. They highlight the continuing importance of maintaining smokefree policies to reduce the perceived desirable attributes of smoking. They also have wider implications in relation to the need to create more positive working environments for young adults, which include working with employers, employer organisations and unions to address the culture of smoking breaks and the working hours and conditions that encourage smoking. Finally, the strong link between smoking and alcohol highlights the challenge of how to decouple further drinking from smoking, perhaps through requiring completely smokefree areas outside bars, a proposal that has found support among social smokers in New Zealand.[21] However, it also presents the opportunity of targeting socioculturally tailored smoking interventions to young adults in bars and clubs, an approach that has produced promising results in the USA.[14 45 46]

**Acknowledgements** The authors would like to thank all of the participants who gave their time and thoughts to the research, as well as colleagues at ScotCen Social Research: Irene Miller and Susan MacLeod for conducting the interviews, Lesley Birse for research and administrative support, and Stephen Hinchliffe and Laura Brown for data management and analysis.

**Contributors** AA and AM conceived the study and designed the methods. AM oversaw participant recruitment. All authors read the transcripts and developed the analysis. HD drafted the manuscript with critical contributions from AA and AM in revised versions.

**Funding** This work was supported by a Cancer Research UK (CRUK) Tobacco Advisory Grant (TAG) award (grant reference number: A21556).

**Competing interests** None declared.

**Patient consent** Not required.

**Ethics approval** Ethical approval was obtained from NatCen (National Centre for Social Research) Research Ethics Committee.

**Provenance and peer review** Not commissioned; externally peer reviewed.

**Data sharing statement** No additional data available.

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
