## [Reviewer comments · BMJ Open]

This paper was submitted to a another journal from BMJ but declined for publication following peer review. The authors addressed the reviewers' comments and submitted the revised paper to BMJ Open. The paper was subsequently accepted for publication at BMJ Open.

(This paper received three reviews from its previous journal but only two reviewers agreed to published their review.)

ARTICLE DETAILS

TITLE (PROVISIONAL)	'Tell them you smoke, you'll get more breaks' – a qualitative study of occupational and social contexts of young adult smoking in Scotland.
AUTHORS	Delaney, Hannah; MacGregor, Andrew; Amos, Amanda

VERSION 1 – REVIEW

REVIEWER	Janet Hoek University of Otago, New Zealand I have met Amanda Amos at several conferences and visited her and her team at the University of Edinburgh in 2015.
REVIEW RETURNED	14-May-2018

GENERAL COMMENTS	I enjoyed reading this interesting and well-written paper and appreciate the opportunity to review it. Recognising the importance of regular smoking uptake (as distinct from age of smoking initiation) has not received detailed attention and the authors correctly note the high smoking prevalence among young adults (an international phenomenon). The background sets a clear context for the research though could draw on more diverse literature as several studies have examined "social" smoking, pairing of smoking and alcohol, and smoking identity positions and stigma management among young adults.1-8 I think it would be worth weaving more international literature into the background, particularly as Ling's work with bar interventions could be drawn upon in the Discussion section. The authors note the difficulty of obtaining a sample and received 15 responses to their invitation letters; it was not clear how many invitation letters they had sent out or why they used mail to contact a group more likely to be using social media or mobile phones? Were alternative contact media available and, if so, why were these not used? Could the authors provide a flowchart illustrating the total initial contacts, gone no addresses, refusals, etc.? Could they also comment on the time period between the potential participants' initial involvement in the SHeS study and the subsequent contact? If potential participants were first interviewed in 2012 and not recontacted until mid to late 2016, many would likely have moved address – to what extent were "uncontactables" a problem? The authors could consider these questions when they discuss their
--

study limitations.

I think it would be helpful to include a copy of the interview guide, life and day grids, and other documents in supplementary files. Table 1 could be expanded to include much of the information in the text and each participant could be given a pseudonym and described in more detail (e.g., age of first cigarette, age at which regular smoking developed, age of first quit attempt, number of quit attempts, date of most recent quit attempt, cpd, time to first cigarette (if obtained – could provide a heaviness of smoking score?). I wondered if the authors could differentiate between age of first-ever cigarette and age of regular smoking (often recognised as at least weekly smoking)?

More information about the age of regular smoking would also provide insights into policy measures – are limits on youth access simply not working (e.g., regular use occurring prior to 16) or do social settings foster smoking (e.g., pairing of smoking and drinking would suggest measures to decouple these behaviours). A detailed table providing this background information on each participant would provide a richer profile of the sample and perhaps support clearer policy insights.

The results are interesting and the findings with respect to workplace smoking offer new insights into how smoking evolves among young adults. There seem to be several sub-themes within this section and I wondered if these could be teased apart a little more? For example, some participants seem to comment on functional benefits – getting more breaks (Sarah and Jamie), which some used to offset boredom – perhaps these ideas could be linked and separated from comments on smoking as a means to bond with workmates (Daniel)? I also wondered if the authors could contrast these “benefits” against the concerns they also report, particularly the idea of being unattractive (smelly) and unprofessional.

The theme on smoking and drinking has been widely reported in the literature; the authors could relocate interpretations of their findings (e.g. bottom para p13) to the discussion section. The section on living circumstances also included diverse ideas and the authors could draw these together a little more. Currently, the section includes comments about smoking as a stress strategy, as a familiar routine and social practice, and as a problematic behaviour (because of effects on others) – these ideas are not always discussed in adjacent paragraphs and some restructuring could improve the flow of this section.

The discussion section summarises the findings but could explore responses to these. For example, the authors could draw more on Pam Ling’s very novel work with bar interventions to explore how the link between smoking and drinking could be broken. Are there opportunities to liaise with major employers to reduce smoking breaks (or provide benefits, as have been recently mooted, to non-smokers who I believe get extra holiday time with some companies because they take fewer breaks)? Or would another response be to extend smokefree perimeters around buildings? Make CBD areas smokefree? Could the authors suggest some implications that future work could test further?

Overall, thank you again for the opportunity to read about this interesting study, which offers thought-provoking findings; I hope my comments are useful to the authors as they develop their MS.

	Minor Suggestions Please add a noun after “this” to ensure the referent is clear. For example, middle para p6 “This has” could be “This change has” or “These disruptions have”; alternatively, final sentence para 1 p5 could be revised to “These measures have significantly...”. References  1. Kingsbury JH, Parks MJ, Amato MS, Boyle RG. Deniers and Admitters: Examining Smoker Identities in a Changing Tobacco Landscape. Nicotine & Tobacco Research. 2016. 2. Song A, Ling P. Social Smoking Among Young Adults: Investigation of Intentions and Attempts to Quit. American Journal of Public Health. 2011;101(7):1291-1296. 3. Hoek J, Maubach N, Stevenson R, Gendall P, Edwards R. Social smokers' management of conflicted identities. Tobacco Control. 2013;22(4):261-265. 4. Marsh L, Cousins K, Connor J, Kypri K, Gray A, Hoek J. The association of smoking with drinking pattern may provide opportunities to reduce smoking among students. Kotuitui. 2016. 5. Gifford H, Tautolo D, Erick S, Hoek J, Edwards R, Gray R. A qualitative analysis of Māori and Pacific smokers' views on informed choice and smoking BMJ Open. 2016. 6. Ling PM, Holmes LM, Jordan JW, Lisha NE, Bibbins-Domingo K. Bars, nightclubs, and cancer prevention: new approaches to reduce young adult cigarette smoking. American journal of preventive medicine. 2017;53(3):S78-S85. 7. Jiang N, Lee YO, Ling PM. Young adult social smokers: Their co-use of tobacco and alcohol, tobacco-related attitudes, and quitting efforts. Preventive Medicine. 2014;69:166-171. 8. McCool J, Hoek J, Edwards R, Thomson G, Gifford H. Crossing the smoking divide for young adults: Expressions of stigma and identity among smokers and non-smokers. Nicotine & Tobacco Research. 2012;15(2):552-556.
--	---

REVIEWER	Kristin Carson-Chahhoud University of South Australia, Australia
REVIEW RETURNED	05-Jun-2018

GENERAL COMMENTS	This is an interesting and well written study. Authors report that participants had diverse smoking histories and educational/occupational histories reducing generalisability. However, given the small sample of 15 participants, it is not clear if data saturation was reached. Can authors clarify how they determined that sufficient information was obtained from the small highly selected sample of responders and consider if this data should be supplemented by recruitment from other sources to improve reliability of findings. The sample is highly selective being responders from SHeS. How many people were invitation letters sent out to and is any information known about the non-responder characteristics compared to responders that may influence the outcomes? Are there potential implications for participant responses due to interviews being conducted by two female researchers opposed to having a male researcher included? Participant employment and smoking status information reported in results could be better presented in a table form. Much of this content is described and presented as you would a qualitative synthesis when the data is actually quantitative (e.g., instead of "some" or "most" used to describe characteristics, place as much of
---

	this information as possible into a characteristics table to improve readability and interpretation of content. Page 10, "six participants had at some point been unemployed". What is the definition for this? For example, if a participant had a hiatus of 3 days between jobs were they considered unemployed? As this definition is the basis for comparisons and contrasts for this analysis, it is important to understand meaning behind the terms used. Some of the above needs to be discussed in the limitations.
--	---

VERSION 1 – AUTHOR RESPONSE

Reviewer: 1

I enjoyed reading this interesting and well-written paper and appreciate the opportunity to review it. Recognising the importance of regular smoking uptake (as distinct from age of smoking initiation) has not received detailed attention and the authors correctly note the high smoking prevalence among young adults (an international phenomenon).

The background sets a clear context for the research though could draw on more diverse literature as several studies have examined “social” smoking, pairing of smoking and alcohol, and smoking identity positions and stigma management among young adults.¹⁻⁸ I think it would be worth weaving more international literature into the background, particularly as Ling’s work with bar interventions could be drawn upon in the Discussion section.

Thank you for the literature suggestions; the Background and Discussion sections have been revised and expanded to draw on this literature at relevant points.

The authors note the difficulty of obtaining a sample and received 15 responses to their invitation letters; it was not clear how many invitation letters they had sent out or why they used mail to contact a group more likely to be using social media or mobile phones? Were alternative contact media available and, if so, why were these not used? Could the authors provide a flowchart illustrating the total initial contacts, gone no addresses, refusals, etc.? Could they also comment on the time period between the potential participants' initial involvement in the SHeS study and the subsequent contact? If potential participants were first interviewed in 2012 and not recontacted until mid to late 2016, many would likely have moved address – to what extent were “uncontactables” a problem? The authors could consider these questions when they discuss their study limitations.

This is now further addressed in the Methods and limitations section in the Discussion. We also reference previous research which highlighted the difficulties of recruiting from this age group.

I think it would be helpful to include a copy of the interview guide, life and day grids, and other documents in supplementary files.

Now added as supplementary files.

Table 1 could be expanded to include much of the information in the text and each participant could be given a pseudonym and described in more detail (e.g., age of first cigarette, age at which regular smoking developed, age of first quit attempt, number of quit attempts, date of most recent quit attempt, cpd, time to first cigarette (if obtained – could provide a heaviness of smoking score?). I wondered if the authors could differentiate between age of first-ever cigarette and age of regular smoking (often recognised as at least weekly smoking)? More information about the age of regular smoking would also provide insights into policy measures – are limits on youth access simply not working (e.g., regular use occurring prior to 16) or do social settings foster smoking (e.g., pairing of smoking and drinking would suggest measures to decouple these behaviours). A detailed table providing this background information on each participant would provide a richer profile of the sample and perhaps support clearer policy insights.

We agree that providing additional information about the participants gives a richer picture of their smoking trajectories. However, not all the information requested was collected but we have included all available additional information in Table 2.

The results are interesting and the findings with respect to workplace smoking offer new insights into how smoking evolves among young adults. There seem to be several sub-themes within this section and I wondered if these could be teased apart a little more? For example, some participants seem to comment on functional benefits – getting more breaks (Sarah and Jamie), which some used to offset boredom – perhaps these ideas could be linked and separated from comments on smoking as a means to bond with workmates (Daniel)? I also wondered if the authors could contrast these “benefits” against the concerns they also report, particularly the idea of being unattractive (smelly) and unprofessional.

Thank you for these suggestions. We have edited the ‘occupational smoking’ section to reflect these comments.

The theme on smoking and drinking has been widely reported in the literature; the authors could relocate interpretations of their findings (e.g. bottom para p13) to the discussion section. The section on living circumstances also included diverse ideas and the authors could draw these together a little more. Currently, the section includes comments about smoking as a stress strategy, as a familiar routine and social practice, and as a problematic behaviour (because of effects on others) – these ideas are not always discussed in adjacent paragraphs and some restructuring could improve the flow of this section.

The ‘drinking’ and ‘living circumstances’ sections have now been revised to reflect these comments.

The discussion section summarises the findings but could explore responses to these. For example, the authors could draw more on Pam Ling’s very novel work with bar interventions to explore how the link between smoking and drinking could be broken. Are there opportunities to liaise with major employers to reduce smoking breaks (or provide benefits, as have been recently mooted, to non-smokers who I believe get extra holiday time with some companies because they take fewer breaks)? Or would another response be to extend smokefree perimeters around buildings? Make CBD areas smokefree? Could the authors suggest some implications that future work could test further?

We were limited by the word count as to how much consideration we could give to the implications and possible future areas for action, but have included several of these points, including reference to Pam Ling’s promising work.

Overall, thank you again for the opportunity to read about this interesting study, which offers thought-provoking findings; I hope my comments are useful to the authors as they develop their MS. **Many thanks for providing such helpful comments.**

Minor Suggestions:

Please add a noun after “this” to ensure the referent is clear. For example, middle para p6 “This has” could be “This change has” or “These disruptions have”; alternatively, final sentence para 1 p5 could be revised to “These measures have significantly...”. **Now updated, please see revised manuscript.**

References

1. Kingsbury JH, Parks MJ, Amato MS, Boyle RG. Deniers and Admitters: Examining Smoker Identities in a Changing Tobacco Landscape. *Nicotine & Tobacco Research*. 2016.
2. Song A, Ling P. Social Smoking Among Young Adults: Investigation of Intentions and Attempts to Quit. *American Journal of Public Health*. 2011;101(7):1291-1296.
3. Hoek J, Maubach N, Stevenson R, Gendall P, Edwards R. Social smokers' management of conflicted identities. *Tobacco Control*. 2013;22(4):261-265.
4. Marsh L, Cousins K, Connor J, Kypri K, Gray A, Hoek J. The association of smoking with drinking pattern may provide opportunities to reduce smoking among students. *Kotuitui*. 2016.
5. Gifford H, Tautolo D, Erick S, Hoek J, Edwards R, Gray R. A qualitative analysis of Māori and Pacific smokers' views on informed choice and smoking *BMJ Open*. 2016.
6. Ling PM, Holmes LM, Jordan JW, Lisha NE, Bibbins-Domingo K. Bars, nightclubs, and cancer prevention: new approaches to reduce young adult cigarette smoking. *American journal of preventive medicine*. 2017;53(3):S78-S85.

7. Jiang N, Lee YO, Ling PM. Young adult social smokers: Their co-use of tobacco and alcohol, tobacco-related attitudes, and quitting efforts. *Preventive Medicine*. 2014;69:166-171.
8. McCool J, Hoek J, Edwards R, Thomson G, Gifford H. Crossing the smoking divide for young adults: Expressions of stigma and identity among smokers and non-smokers. *Nicotine & Tobacco Research*. 2012;15(2):552-556.

These papers have been considered and incorporated into the revised manuscript to enhance the Background and Discussion sections.

Reviewer: 2

This is an interesting and well written study.

Many thanks for your helpful comments.

Authors report that participants had diverse smoking histories and educational/occupational histories reducing generalisability. However, given the small sample of 15 participants, it is not clear if data saturation was reached. Can authors clarify how they determined that sufficient information was obtained from the small highly selected sample of responders and consider if this data should be supplemented by recruitment from other sources to improve reliability of findings.

While we cannot say that data saturation was reached, the sample was diverse yet the participants discussed similar experiences and their accounts included several overlapping themes. More information has been provided about the recruitment process, which was from a nationally representative survey.

The sample is highly selective being responders from SHeS. How many people were invitation letters sent out to and is any information known about the non-responder characteristics compared to responders that may influence the outcomes?

SHeS is representative at the general population level. The timelag between the SHeS survey and the interviews makes it difficult to answer the above point, for instance we discovered during the study that; people moved between employment and NEET, changed living circumstances and smoking status since taking part in SHeS. The recruitment and sampling has been further addressed in the Methods section.

Are there potential implications for participant responses due to interviews being conducted by two female researchers opposed to having a male researcher included?

Two highly experienced interviewers carried out the data collection and elicited a wealth of information from both the male and female participants.

Participant employment and smoking status information reported in results could be better presented in a table form. Much of this content is described and presented as you would a qualitative synthesis when the data is actually quantitative (e.g., instead of "some" or "most" used to describe characteristics, place as much of this information as possible into a characteristics table to improve readability and interpretation of content.

This information is included in Table 1 and also now in an additional Table 2.

Page 10, "six participants had at some point been unemployed". What is the definition for this? For example, if a participant had a hiatus of 3 days between jobs were they considered unemployed? As this definition is the basis for comparisons and contrasts for this analysis, it is important to understand meaning behind the terms used. Some of the above needs to be discussed in the limitations.

As this was a qualitative study there were no pre-set criteria for this, but it was clear that when respondents spoke of times that they were out of work, this was about periods of at least several weeks (clarification now included in revised manuscript).

VERSION 2 – REVIEW

REVIEWER	Kristin Carson-Chahhoud University of South Australia, Australia
REVIEW RETURNED	10-Oct-2018

GENERAL COMMENTS	Some mention of data saturation needs to be included, given this is a qualitative study where you are trying to understand smoking within a particular context. There also needs to be a dedicated and clear limitations section within the discussion. Otherwise, everything looks good.
---

VERSION 2 – AUTHOR RESPONSE

Reviewer: 2

Some mention of data saturation needs to be included, given this is a qualitative study where you are trying to understand smoking within a particular context. **The Methods section has now been revised to reflect this comment.**

There also needs to be a dedicated and clear limitations section within the discussion. Otherwise, everything looks good. **Discussion section revised to include limitations.**